# Curcumin Alleviates IUGR Jejunum Damage by Increasing Antioxidant Capacity through Nrf2/Keap1 Pathway in Growing Pigs

**DOI:** 10.3390/ani10010041

**Published:** 2019-12-24

**Authors:** Enfa Yan, Jiaqi Zhang, Hongli Han, Jiamin Wu, Zhending Gan, Chengheng Wei, Lili Zhang, Chao Wang, Tian Wang

**Affiliations:** College of Animal Science and Technology, Nanjing Agricultural University, No. 6, Tongwei Road, Xuanwu District, Nanjing 210095, China; 2018805112@njau.edu.cn (E.Y.); 2019105063@njau.edu.cn (J.Z.); 2018105060@njau.edu.cn (H.H.); 2018105066@njau.edu.cn (J.W.); 2017105067@njau.edu.cn (Z.G.); 2019805114@njau.edu.cn (C.W.); zhanglili@njau.edu.cn (L.Z.)

**Keywords:** IUGR, curcumin, jejunum, antioxidant, apoptosis, immune system, tight junction

## Abstract

**Simple Summary:**

Intrauterine growth retardation (IUGR) is usually defined as fetal growth below the 10th percentile for gestational age and results in impaired growth and development of the fetus and/or its organs during gestation. IUGR not only has a permanent effect on postnatal growth and long-term health, but also results in high fetal mortality and morbidity. Recent results have demonstrated that IUGR can cause jejunum damage in piglets. The jejunum is not only the main organ for the digestion and absorption of nutrients, but also an immune organ in the body. However, few effective methods to alleviate jejunum damage and oxidative stress in IUGR pigs have been found. In recent years, studies have found that curcumin (CUR) may be an effective and safe feed additive for regulating oxidative stress in the body. Our results indicated that dietary added 200 mg/kg curcumin to the basal diet can increase the antioxidant capacity of the IUGR growing pigs, jejunum and alleviate the damage in jejunum of the IUGR growing pigs. Therefore, the use of curcumin as a feed additive has certain economic value.

**Abstract:**

The purpose of this study was to explore the effects of curcumin on IUGR jejunum damage. A total of 24 IUGR and 12 normal-birth weight (NBW) female crossbred (Duroc × Landrace × Large White) piglets were randomly assigned into three groups at weaning (26 days): IUGR group, NBW group, and IUGR + CUR group, which were fed diets containing 0 mg/kg (NBW), 0 mg/kg (IUGR) and 200 mg/kg (IUGR + CUR) curcumin from 26 to 115 days of age. Results showed that dietary supplementation with 200 mg/kg curcumin significantly increased the total superoxide dismutase (T-SOD) activity and decreased the malondialdehyde (MDA) content in the jejunum of IUGR pigs (*p* < 0.05). Results of real-time PCR showed that the IUGR + CUR group significantly increased the gene expression of NF-E2-related factor 2 (*Nrf2*) (*p* < 0.05), and increased the glutamate-cysteine ligase catalytic subunit (*GCLC*), superoxide dismutase 1 (*SOD1*), glutamate-cysteine ligase modifier subunit (*GCLM*), and NAD(P)H quinone dehydrogenase 1 (*NQO1*) mRNA expression compared with the IUGR group (*p* < 0.05). Western blot results showed that dietary supplementation with 200 mg/kg curcumin significantly increased the protein levels of Nrf2 and NQO1. Compared with the IUGR group, pigs in IUGR + CUR group showed significantly decreased the levels of tumor necrosis factor-α (*TNFα*), interleukin-6 (*IL-6*), and interferon gamma (*IFNγ*) (*p* < 0.05), and increased the interleukin-2 (*IL-2*) level (*p* < 0.05). Dietary supplementation with 200 mg/kg curcumin significantly reduced cysteinyl aspartate specific proteinase 3 (*caspase3*), BCL2-associated X protein (*bax*), B-cellCLL/lymphoma 2 (*bcl2*), and heat-shock protein 70 (*hsp70*) mRNA expression, and increased occludin (*ocln*) mRNA expression (*p* < 0.05). In conclusion, dietary supplementation with 200 mg/kg curcumin can alleviate jejunum damage in IUGR growing pigs, through Nrf2/Keap1 pathway.

## 1. Introduction

Intrauterine growth restriction (IUGR), characterized by impaired development of the fetus during pregnancy, is a major threat in animal production because of the harmful effect on postnatal growth performance and health status [1]. A piglet was defined as suffering intrauterine growth restriction when its birth weight was two standard deviations below the mean birth weight of the total population, or the neonatal birth weight of less than 10% with gestational age of normal weight [2,3]. There is increasing evidence showing that IUGR may also have long-term consequences on the offspring resulting in high neonatal morbidity and mortality [4]. The jejunum is not only the main place for the digestion and absorption of nutrients, but also the immune organ in the body [5]. Recent results have demonstrated that IUGR can cause jejunum damage in piglets at 21 and 50 days [6,7,8]. As for the IUGR programming consequences, IUGR might affect the jejunum function of growing pigs, such as immunization, intestinal barrier, proliferation, and apoptosis, but few relevant reports have been reported so far.

IUGR can cause oxidative stress [9]. Previous research has shown that IUGR pigs exhibited decreases in activities of total antioxidant capacity and total superoxide dismutase and increased levels of malondialdehyde (MDA) and protein carbonyl (PC) [10]. The effect of IUGR on jejunum function is closely related to oxidative function. For example, previous research proved that imbalance of antioxidant homeostasis is an important factor contributing to intestinal mucosal injury induced by IUGR [8]. However, few effective methods to alleviate jejunum damage and oxidative stress in IUGR pigs have been found.

In recent years, studies have found that curcumin may be an effective and safe feed additive for regulating oxidative stress in the body. Curcumin is the main ingredient of the spice turmeric (*Curcuma longa*), and has been administered as traditional medicine in Asia, mainly in China, India, and Iran [11]. The yellow-pigmented fraction of turmeric contains curcuminoids, which are chemically related to its principal ingredient, curcumin. The major curcuminoids present in turmeric are demethoxycurcumin bisdemethoxycurcumin and the recently identified cyclocurcumin [12]. Studies indicate that functional groups associated with curcumin chemical structure including bis-α, β-unsaturated, and β-diketone, two methoxy groups, two phenolic hydroxy groups, and two double-conjugated bonds might play an essential role in antiproliferative and anti-inflammatory activities assigned to curcumin [13].

Curcumin also downregulates cyclin D1, cyclin E, and murine double minute 2 (MDM2); and upregulates tumor protein p53 (p53) [14]. Curcumin has been known for decades as an antioxidative and anti-inflammatory substance [11]. What is more, curcumin has been proven to be a bifunctional antioxidant that exerts antioxidant activity both directly and indirectly by scavenging reactive oxygen species (ROS) and inducing an antioxidant response [15]. Previous studies concluded that curcumin can improve oxidant stability of muscle in broilers [16]. Some research suggested that curcumin is effective in the oxidant–antioxidant imbalance [17]. We have studied the antioxidant function of the jejunum and determined the index of partial immunization, apoptotic and intestinal tight junction protein. We speculate that curcumin may regulate the jejunum function of IUGR growing pigs through the Nrf2/Keap1 signaling pathway. Our trials are designed to provide a reliable reference for the prevention and treatment of intestinal function injury in animals.

## 2. Materials and Methods

### 2.1. Ethical Statement

All the procedures were carried out in accordance with the Chinese Guidelines for Animal Welfare and Experimental Protocol, and were approved by the Institutional Animal Care and Use Committee of Nanjing Agricultural University, China (NJAU-CAST-2015-098).

### 2.2. Animal and Experimental Design

At the time of parturition (114 days gestation), a total of 24 IUGR and 12 normal-birth weight (NBW) female crossbred (Duroc × Landrace × Large White) piglets chosen from 12 sows were used in this study. Two IUGR and one NBW piglets were selected from each litter. At weaning (26 days), all piglets were randomly assigned into 3 groups (each group with 12 piglets). The NBW group and the IUGR group were fed the basal diet, and the IUGR + CUR group was fed the basal diet supplemented with 200 mg/kg curcumin (Guangdong Kehu Biotechnology Research and Development Center; Guangzhou, China, purity ≥ 98%). The effective content of curcumin (200 mg/kg) was selected according to previous study [18].

The pigs were fed under the same conditions of temperature, humidity, and ventilation, and pigs had water and food ad libitum. We cleaned the pig house every day and cleaned the drinking water pipes regularly, and ventilated in time. The immunization procedure was carried out as usual.

At the end of feeding trial (115 days old), 6 pigs (fasting for 12 h) were selected from each group, euthanized by electrical stunning, and exsanguinated. The jejunal mucosa samples were frozen in liquid nitrogen and stored at −80 °C for further determination.

### 2.3. Assay of Antioxidant Index in the Jejunal Mucosa

The jejunal mucosa samples were homogenized in ice-cold 0.90% sodium chloride buffer (*w*:*v*, 1:9). The supernatant was obtained by centrifugation at 5000× *g* for 10 min at 4 °C, which was used for the further determination. MDA (kit number: A003-1-1), total superoxide dismutase (T-SOD, kit number: A001-1-1), glutathione peroxide enzyme (GPx, kit number: A005-1-1) and total antioxidant capacity (T-AOC, kit number: A015-1-1, FRAP) were determined with corresponding commercial kits purchased from Nanjing Jiancheng Bioengineering Institute (Nanjing, China). The activities of T-SOD, T-AOC, and GPx were expressed as units (U) per milligram of protein. The concentration of MDA was expressed as nanomoles per milligram of protein.

### 2.4. Total RNA Isolation and mRNA Quantification

Total RNA was isolated using Trizol Reagent (Vazyme, Nanjing, China) from snap-frozen jejunal mucosa using the manufacturer’s protocol. The RNA integrity was checked on a 1% ethidium bromide-stained 1.4% agarose formaldehyde gel. Reverse transcription was performed using a commercial kit (PrimeScript RT Reagent Kit; TaKaRa Biotechnology, Dalian, China). The RNA concentration and purity were calculated from the value of OD_260_/OD_280_ (2.1 > ratio > 1.8) using a NanoDrop ND-2000 UV spectrophotometer (NanoDrop Technologies, Wilmington, DE, USA). Total RNA (1 µg) was reverse-transcribed into cDNA using the PrimeScript RT Reagent Kit (TaKaRa Biotechnology, Dalian, China) according to the manufacturer’s guidelines. Quantitative real-time polymerase chain reaction (qRT-PCR) was performed on an ABI StepOnePlus^TM^ Real-Time PCR System (Applied Biosystems, Grand Island, NY, USA). The cDNA was stored at −20 °C for further determination. The sequence of primers used in this experiment are shown in Table 1. The SYBR Green PCR reaction system was 10 μL in total, which was composed of 5 μL ChamQ SYBR qPCR Master Mix (2×), 0.2 μL forward and reverse primers, 0.2 μL ROX Reference Dye 2 (50×), 1 μL cDNA, and 3.6 μL ddH_2_O. The relative mRNA expression of target genes were calculated using the 2^−ΔΔCt^ method as previously reported [19]. Primer sequences are shown in Table 1.

### 2.5. Western Blot Analysis

The protein of the jejunal mucosa was extracted with a strong cell lysate containing Phenylmethanesulfonyl fluoride (PMSF; Beyotime Institute of Biotechnology, Nantong, China), and the protein concentration in the supernatant was determined by a BCA kit (Beyotime Institute of Biotechnology, Nantong, China) The adjusted protein concentration was diluted to 10 μg/μL with a cell lysate containing PMSF, then 5 times sodium dodecyl sulfate (SDS) supernatant gel buffer was added, and then protein was denatured at 99 °C on a PCR machine. A suitable concentration of SDS-PAGE gel was prepared according to the molecular weight of the detected protein; 10 μL of the protein sample and the pre-stained protein marker were directly loaded into the SDS-PAGE gel-loading well; then electrophoresis was performed. After the end of the electrophoresis, the gel was cut to an appropriate size, and a similarly sized polyvinylidene fluoride (PVDF) membrane (soaked for at least 15 s in methanol) was cut; then, the membrane was transferred in transfer solution. After the membrane was transferred, it was blocked with a 5% skim milk powder solution. Then, the membranes were incubated overnight with primary antibodies: Nrf2 (1:1000; Proteintech; Rosemont, IL, USA), keap1 (1:1000; Proteintech; Rosemont, IL, USA), NQO1 (1:1000; Proteintech; Rosemont, IL, USA), α-Tubulin (1:1000; Proteintech; Rosemont, IL, USA) at 4 °C. The membranes were washed in TBST three times and were processed with secondary antibody (1:5000; Proteintech; horseradish-peroxidase-conjugated goat anti-rabbit immunoglobulin G (IgG); Rosemont, IL, USA) for 60 min at room temperature. The blots were developed using an enhanced Chemiluminescence reagents (Merck Millipore, Darmstadt, Germany) followed by autoradiography. Images were recorded using a Luminescent Image Analyzer LAS-4000 system (Fujifilm, Tokyo, Japan) and were quantified by Image-Pro Plus 6.0 (Media Cybernetics, MD, USA). α-tubulin was used as the internal standard to normalize the signals.

### 2.6. Statistical Analysis

The results of the test were calculated by Excel 2016. Data were expressed as means with SEM (standard error of mean) and analyzed by one-way ANOVA using the SPSS 25.0 software (SPSS, Inc., Chicago, IL, USA). Multiple comparisons were performed by Duncan’s method. The data were expressed as mean ± standard error; when *p* < 0.05, the difference was considered significant.

## 3. Results

### 3.1. Antioxidant Index in the Jejunal Mucosa

Effects of dietary curcumin on antioxidant capacity of jejunal mucosa in IUGR growing pigs are shown in Table 2. Results showed that pigs in IUGR group had higher MDA content (*p* < 0.05) than pigs in NBW group. Compared with the IUGR group, dietary added 200 mg/kg of curcumin significantly decreased MDA content (*p* < 0.05), and increased T-SOD (*p* < 0.05) activity in jejunal mucosa. In addition, there were no significant differences in antioxidant enzyme activities of GPx and T-AOC among the three groups (*p* > 0.05).

### 3.2. Antioxidant Enzyme Gene Expression in the Jejunal Mucosa

As shown in Figure 1, IUGR pigs had lower mRNA expression of *Nrf2*, *SOD1*, and *GCLC* as compared with NBW pigs (*p* < 0.05). Compared with IUGR group, the group supplemented with 200 mg/kg of curcumin significantly increased mRNA expression of *Nrf2*, *SOD1*, *GCLM*, *GCLC*, and *NQO1* in jejunal mucosa (*p* < 0.05). There were no significant differences in mRNA expression of *Keap1*, *HO-1*, and *CAT* in jejunal mucosa of pigs between IUGR + CUR and NBW groups (*p* > 0.05).

### 3.3. Hsp70 Expression in the Jejunal Mucosa

As shown in Figure 2, pigs in IUGR group had a higher mRNA expression of *hsp70* compared with pigs in the IUGR + CUR group (*p* < 0.05). There were no significant differences in *hsp70* mRNA expression between IUGR and NBW groups (*p* > 0.05). There were no significant differences in *hsp70* mRNA expression in jejunal mucosa between IUGR + CUR and NBW groups (*p* > 0.05).

### 3.4. Immune-Related Gene Expression in the Jejunal Mucosa

As shown in Figure 3, pigs in IUGR group had a higher mRNA expression of *TNFα*, *IL-6*, and *IFNγ* as compared with pigs in the NBW group (*p* < 0.05). Compared with the IUGR group, dietary supplementation with 200 mg/kg of curcumin significantly decreased mRNA expression of *TNFα*, *IL-6*, and *IFNγ*, and increased mRNA expression of *IL-2* in jejunal mucosa (*p* < 0.05). There were no significant differences in mRNA expression of *IL-1β*, *TNFα*, *IL-6*, *IL-2*, and *IFNγ* between the IUGR + CUR and NBW groups (*p* > 0.05).

### 3.5. Apoptosi- Related Gene Expression in the Jejunal Mucosa

As shown in Figure 4, pigs in the IUGR group had higher mRNA expression of *caspase3* and *bcl2* as compared with pigs in the NBW group (*p* < 0.05). Compared with the IUGR group, dietary supplementation with 200 mg/kg of curcumin significantly decreased mRNA expression of *caspase3*, *bax*, and *bcl2* in jejunal mucosa (*p* < 0.05). There were no significant differences in mRNA expression of *caspase9*, *caspase3*, *bax*, and *bcl2* in jejunal mucosa between IUGR + CUR and NBW groups (*p* > 0.05). The mRNA expression of *p53* had no significant differences among the IUGR, NBW, and IUGR + CUR groups (*p* > 0.05).

### 3.6. Tight Junction-Related Gene Expression in the Jejunal Mucosa

As shown in Figure 5, IUGR pigs had a lower *ocln* mRNA expression as compared with NBW pigs (*p* < 0.05) in jejunal mucosa. Compared with IUGR group, dietary supplementation with 200 mg/kg of curcumin significantly increased *ocln* mRNA expression (*p* < 0.05). There were no significant differences in *ocln* mRNA expression between IUGR + CUR and NBW groups (*p* > 0.05). The *ZO-1* mRNA expression had no significant differences among the IUGR, NBW, and IUGR + CUR groups (*p* > 0.05).

### 3.7. Protein Expression of Keap1/Nrf2 Signal Pathway in Jejunal Mucosa

Effects of curcumin on protein expression of Nrf2/Keap1 signal pathway in jejunal mucosa are shown in Figure 6. As the Western blot results show, the expression of Nrf2 and NQO1 protein in the IUGR group was decreased (*p* < 0.05) in the jejunum compared with those in the NBW group (Figure 6). No difference (*p* > 0.05) was found between the NBW group and the IUGR + CUR group. The expression of Keap1 protein had no difference among three groups.

## 4. Discussion

Nrf2/Keap1 is the key signaling pathway of antioxidant damage in organism [20]. The transcription factor Nrf2 regulates the basal and inducible expression of numerous detoxifying and antioxidant genes [21]. The nuclear transcription factor Nrf2 is an important transcription factor in the antioxidant system. Under physiological conditions, it binds to the cytosolic chaperone Keap1 to maintain the activity relatively inhibited. Upon exposure to oxidative stressors, Nrf2 evades Keap1-mediated inhibition, enters the nucleus, binds to antioxidants, and activates a range of downstream antioxidant enzymes, such as HO-1, SOD, and NQO1 [21]. Nrf2 regulates gene expression, such as *GCLC* and *GCLM* that defend various tissues against diverse electrophilic stressors and oxidative insults, thus protecting the organism from disorders that are caused or exacerbated by such stressors [22]. Oxidative stress can increase the risk of metabolic syndrome in IUGR infants as adults [23]. Our results indicate that IUGR reduces the expression of Nrf2 mRNA and protein in the jejunum of growing pigs. However, dietary supplementation with 200 mg/kg curcumin can increase the expression of Nrf2 mRNA and protein, which could explain that curcumin maybe promote the expression of Nrf2.

Adding curcumin to the diet of hyperuricemia mice can restore normal antioxidant enzyme activities (SOD, GPx) and reduce the accumulation of MDA in serum [24]. Previous research has found that IUGR impaired antioxidant defense system and increased oxidative stress by decreasing the activities of antioxidant enzymes, such as GPx and Cu-Zn SOD [25]. SOD is an important antioxidant enzyme in the body, which can effectively maintain the balance of free radicals in the body [26]. From our results, it was found that the addition of 200 mg/kg curcumin to the diet increased the T-SOD activity and pigs in IUGR + CUR group had significantly higher mRNA expression of SOD1 than those of IUGR group, which is similar to the previous study [27]. MDA were considered as a marker of lipid peroxidation [28]. IUGR can increase the content of MDA in pigs’ jejunal mucosa [29], which is consistent with our experimental results, and dietary supplementation with 200 mg/kg curcumin can reduce the content of IUGR jejunal MDA in growing pigs. These further indicate that curcumin, as an antioxidant, can improve the antioxidant capacity of growing pigs^,^ jejunum.

Hsp70 has anti-apoptosis role and enhances cellular immune function [30]. Oxidative stress can damage the intestinal mucosa of piglets and induce the expression of hsp70 [31]. A previous study found that IUGR offspring exhibited increased levels of heat-shock proteins in the jejunum, which provide a line of evidence for the presence of oxidative stress during postnatal life [32]. Our experiments also demonstrated that curcumin can significantly reduce the *hsp70* mRNA expression in IUGR pigs, which further proves that hsp70 has a protective effect on the intestinal mucosa.

The intestinal tract plays an important role in maintaining epithelial immune stability, and it can form a natural physical barrier and express a large number of antimicrobial peptides to prevent contact between intestinal microbes and immune cells [33]. Endogenous Nrf2 is involved in inflammatory response and wound repair [34]. Braun’s experiment found that Nrf2 was highly expressed in post-traumatic epithelial cells and inflammatory due to Nrf2 deficiency, TNFα expression continued to increase, and the wound epithelial repair period was prolonged. Braun therefore proposes Nrf2 as a new concept for promoting inflammation recovery media [34]. IUGR can damage the jejunal mucosa of newborn piglets, develop jejunum and mesenteric lymph nodes, damage the jejunal mucosal mechanical barrier, reduce the number of immune cells in the jejunal mucosa, and decrease the ability of the jejunal mucosa to secrete cytokines [35]. TNF-α is the primary mediator of inflammation and activates the inflammatory response of the innate immune system, including inducing the production of cytokines such as IL-6 [36]. A previous study has shown that the protective effect of curcumin on necrotizing enterocolitis is accompanied by the weakening of pro-inflammatory effect and the relative enhancement of inflammatory inhibition [37]. In our study, curcumin can significantly inhibit the *TNF-α*, *INF-γ*, and *IL-6* mRNA expression, and increase *IL-2* mRNA expression in jejunal mucosa, which are similar to Xun Wenjuan team’s conclusions [38]. Therefore, we speculate that curcumin may alleviate jejunal inflammation in IUGR growing pigs through the Nrf2 pathway.

In a model of TNFα treatment-induced apoptosis, it was demonstrated that Nrf2-deficient thymocytes die quickly and Nrf2-deficient mice show severe hepatitis, and Nrf2 can reduce the sensitivity of cells to apoptotic signals by regulating cell oxidative balance [39]. Normal apoptosis of jejunum epithelial cells also plays an important role in the growth and development of the jejunum tract and the regulation of immune function. The jejunal mucosal barrier is composed of a layer of jejunum epithelial cells; under normal conditions, the rate of apoptosis and proliferation of jejunum epithelial cells maintains a relatively balanced state [5]. However, when the body produces inflammation or endotoxin stimulation, the jejunum homeostasis is unbalanced, and jejunum epithelial cells accelerate apoptosis, resulting in jejunum epithelial barrier damage [5]. The balance between pro-apoptotic and anti-apoptotic genes in the bcl2 family proteins determines cell survival. Bax as a pro-apoptotic gene can form a dimer by binding to the anti-apoptotic gene bcl2, thereby promoting apoptosis [40,41]. Previous research found that curcumin can inhibit the occurrence of apoptosis in jejunum epithelial cells play a role in relieving jejunum damage [5]. Similarly, our results showed that dietary supplementation with 200 mg/kg curcumin significantly reduced the *bax*, *bcl2*, and *caspase* mRNA expression, which may be one of the signs that curcumin relieves jejunum damage.

Tight junction is the most important connection of intestinal epithelial cells, which is critical in maintaining the integrity of intestinal epithelial cell structure, protecting the intestinal barrier function, preventing bacteria endotoxin, and toxic macromolecules entry into the body [42]. Tight junction is an important component of the jejunal mucosal mechanical barrier, including ocln, cldn, ZO-1, and connective adhesion molecules [43]. ZO-1 plays an important role in maintaining and regulating the integrity of the tight junction complex. Ocln, as one of the most important structural proteins in tight junctions, binds to proteins such as ZO-1 to form a tightly linked backbone [44]. Once ocln enters a tight junction, it will reduce the permeability of the membrane to which it is attached, thereby protecting the jejunal mucosal barrier [43]. From our results, it was found that dietary supplementation with 200 mg/kg curcumin increased the *ocln* mRNA expression in the IUGR growing pigs, while *ZO-1* did not appear to be significantly up-regulated. At present, it is still unclear how curcumin regulates the expression or function of the *ZO-1* mRNA. We speculate that curcumin can effectively promote the ocln the formation of tight connection structure between the cells, thereby reducing the permeability of intestine, and enhancing the intestinal cell barrier function.

## 5. Conclusions

In conclusion, IUGR can cause oxidative stress in the jejunum of growing pigs, destroy the antioxidant defense system, increase apoptosis, and reduce jejunum immunity. The addition of 200 mg/kg curcumin to the diet may alleviate the jejunum oxidative stress of the IUGR growing pigs and improve the jejunum antioxidant function through the Nrf2/Keap1 pathway, thereby further improving the jejunum immune function of IUGR pigs, and improving jejunal tight junction, but the relevant mechanisms are need further research. The beneficial effect of curcumin requires further development, and our findings may be helpful in exploiting a new healthcare product and provide a new basis for alleviating IUGR offspring jejunum oxidative damage.

## Figures and Tables

**Figure 1 animals-10-00041-f001:**
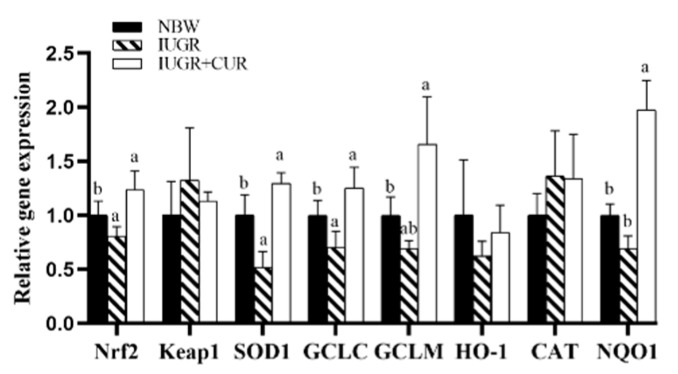
Effects of curcumin on antioxidant-related gene expression: NF-E2-related factor 2 (*Nrf2*), kelch like ECH associated protein 1 (*Keap1*), superoxide dismutase 1 (*SOD1*), glutamate-cysteine ligase catalytic subunit (*GCLC*), glutamate-cysteine ligase modifier subunit (*GCLM*), heme oxygenase-1 (*HO-1*), catalase (*CAT*) and NAD(P)H quinone dehydrogenase 1 (*NQO1*) in jejunal mucosa. Data are normalized to the normal-birth weight (NBW) group and expressed as mean ± SE (n = 6); a, b means that the same parameter with different superscripts are significantly different (*p* < 0.05).

**Figure 2 animals-10-00041-f002:**
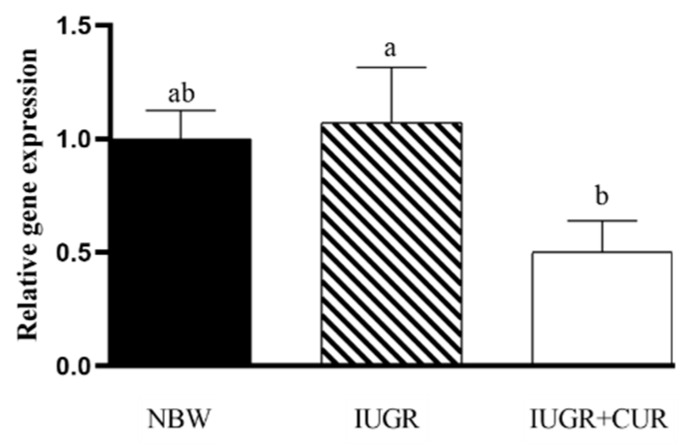
Effects of curcumin on heat-shock protein 70 (*hsp70*) gene expression in jejunal mucosa. Data are normalized to the NBW group and expressed as mean ± SE (n = 6); a, b means that the same parameter with different superscripts are significantly different (*p* < 0.05). NBW: normal-birth weight group; IUGR: intrauterine growth retardation group; IUGR+CUR: intrauterine growth retardation+curcumin group.

**Figure 3 animals-10-00041-f003:**
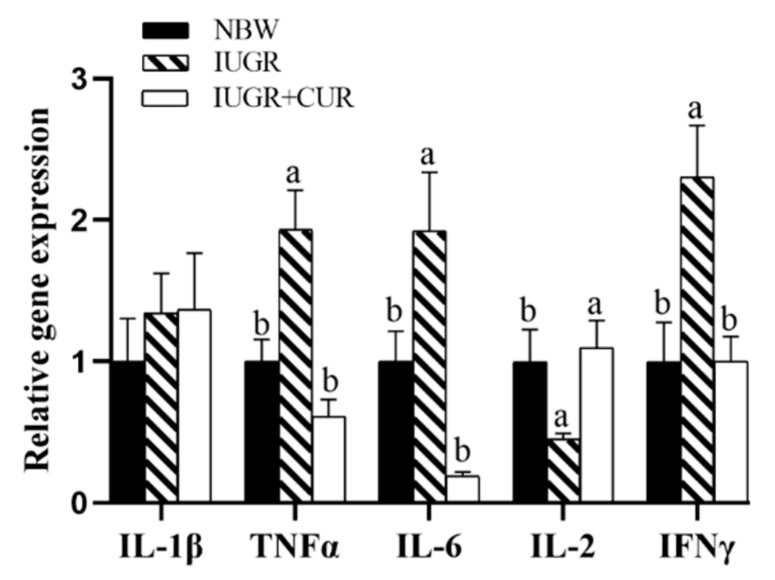
Effects of curcumin on immune-related gene expression: interleukin-1β (*IL-1β*), tumor necrosis factor-α (*TNFα*), interleukin-6 (*IL-6*), interleukin-2 (*IL-2*) and Interferon gamma (*IFNγ*) in jejunal mucosa. Data are normalized to the NBW group and expressed as mean ± SE (n = 6); a, b means that the same parameter with different superscripts are significantly different (*p* < 0.05).

**Figure 4 animals-10-00041-f004:**
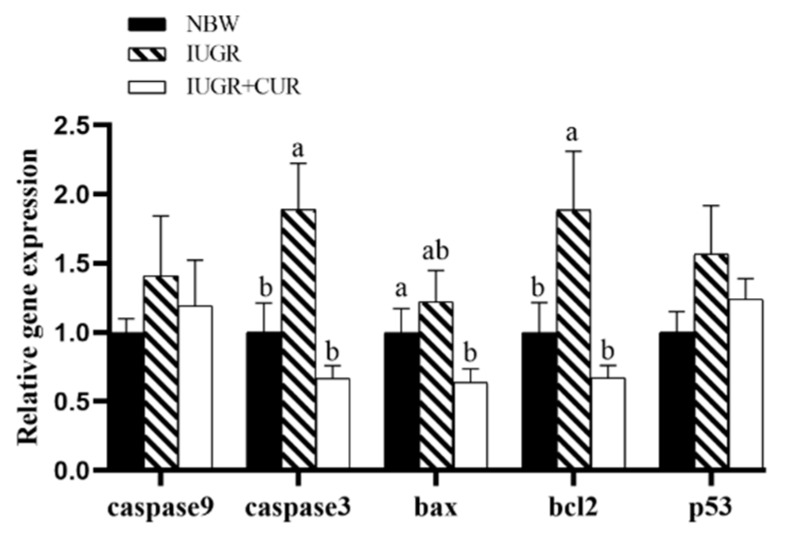
Effects of curcumin on apoptosis-related gene expression: cysteinyl aspartate specific proteinase9 (*caspase9*), cysteinyl aspartate specific proteinase3 (*caspase3*), BCL2-associated X protein (*bax*), tumor protein p53 (*p53*) and B-cellCLL/lymphoma 2 (*bcl2*) in jejunal mucosa. Data are normalized to the NBW group and expressed as mean ± SE (n = 6); a, b means that the same parameter with different superscripts are significantly different (*p* < 0.05).

**Figure 5 animals-10-00041-f005:**
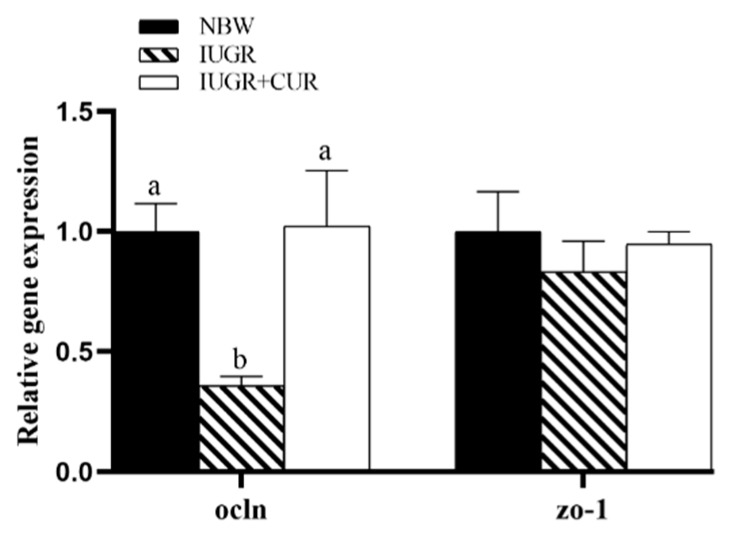
Effects of curcumin on occludin (*ocln*) and tight junction protein 1 (*ZO-1*) gene expression in jejunal mucosa. Data are normalized to the NBW group and expressed as mean ± SE (n = 6); a, b means that the same parameter with different superscripts are significantly different (*p* < 0.05).

**Figure 6 animals-10-00041-f006:**
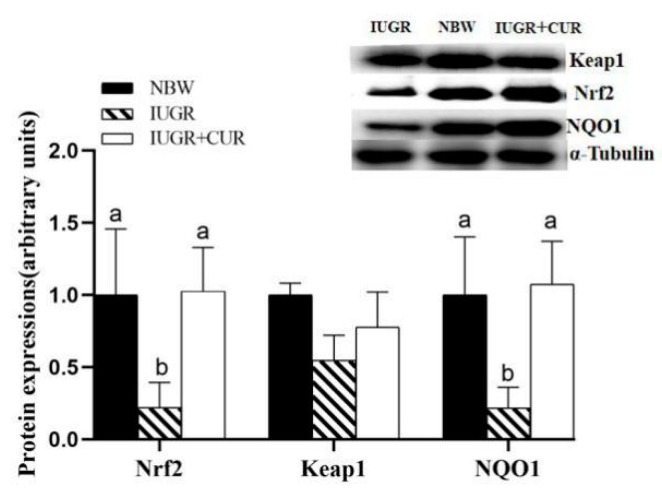
Effects of curcumin on protein expression of Keap1/Nrf2 signal pathway: NF-E2-related factor 2 (Nrf2), kelch like ECH associated protein 1 (Keap1) and NAD(P)H quinone dehydrogenase 1 (NQO1) in jejunal mucosa. Data are expressed relative to α-Tubulin and normalized to the NBW group and expressed as mean ± SE (n = 4); a, b means that the same parameter with different superscripts are significantly different (*p* < 0.05).

**Table 1 animals-10-00041-t001:** Primer sequences used in quantitative real-time PCR assays.

Gene	Accession No	Sequence (5′ to 3′)	Product Length (bp)
*β-actin*	XM_003124280.4	F: CACGCCATCCTGCGTCTGGA R: AGCACCGTGTTGGCGTAGAG	380
*hsp70*	X68213	F: GCCCTGAATCCGCAGAATA R: TCCCCACGGTAGGAAACG	281
*p53*	NM_213824.3	F: CACTGGATGGCGAGTATTTCAC R: CTTAGACTTCAGGTGGCTGGA	152
*IFNγ*	AY188090.1	F: TCAGCTTTGCGTGACTTTGTG R: GCTCTCTGGCCTTGGAACAT	251
*ocln*	NM_001163647.2	F: ATGGCTGCCTTCTGCTTCAT R: TCACTTTCCCGTTGGACGAG	75
*caspase3*	NM_214131.1	F: ATAATTCAGGCCTGCCGAAG R: TGTTTCAGCGCTGCACAAAG	200
*caspase9*	XM_003127618.	F: ACCCCTGACATGATCGAGGA R: CTGGCTTGAGTTCCACTGGT	256
*bax*	XM_003127290.4	F: AGCATGCGGCCTCTATTTGA R: GGCCCGTGGACTTCACTTAT	200
*IL-2*	NM_213861.1	F: TGCACTAACCCTTGCACTCA R: GCAATGGCTCCAGTTGTTTCT	83
*TNFα*	NM_214022.1	F: ATCGGCCCCCAGAAGGAAGAG R: GATGGCAGAGAGGAGGTTGAC	351
*IL-6*	NM_214399.1	F: AAATGTCGAGGCTGTGCAGA R: CTCAGGCTGAACTGCAGGAA	207
*IL-1β*	NM_214029.1	F: TGCCAGCTATGAGCCACTTCC R: TGACGGGTCTCGAATGATGCT	337
*Nrf2*	NM_001114671.1	F: GACAAACCGCCTCAACTCAG R: GTCTCCACGTCGTAGCGTTC	183
*Keap1*	XM_021076667.1	F: CGTGGAGACAGAAACGTGGA R: CAATCTGCTTCCGACAGGGT	239
*NQO1*	NM_001159613.1	F: GATCATACTGGCCCACTCCG R: GAGCAGTCTCGGCAGGATAC	200
*GCLC*	XM_003482164.4	F: GGCGACGAGGTGGAATACAT R: GTTTGGGTTTGTCCTTTCCCC	123
*GCLM*	XM_001926378.4	F: GCATCTACAGCCTTACTGGGA R: GTTAAATCGGGCGGCATCAC	180
*HO-1*	NM_001004027.1	F: CAAGCAGAAAATCCTCGAAG R: GCTGAGTGTCAGGACCCATC	241
*SOD1*	NM_001190422.1	F: CATTCCATCATTGGCCGCAC R: TTACACCACAGGCCAAACGA	118
*ZO-1*	XM005659811.1	F-ACCCCCTACATGCTGACTCT R-TGGCTGCTTCAAGACATGGT	167
*CAT*	XM_021081498.1	F: AGCTTTGCCCTTGCACAAAC R: ACATCCTGAACAAGAAGGGGC	119

*β-actin*, beta-actin; *hsp70*, heat-shock protein 70; *p53*, tumor protein p53; *IFNγ*, Interferon gamma; *ocln*, occludin; *Bcl-2*, B-cellCLL/lymphoma 2; *caspase3*, cysteinyl aspartate specific proteinase 3; *caspase9*, cysteinyl aspartate specific proteinase 9; *bax*, BCL2-associated X protein; *IL-2*, interleukin-2; *TNF-α*, tumor necrosis factor-α; *IL-6*, interleukin-6; *IL-1β*, interleukin-1β; *Nrf2*, NF-E2-related factor 2; *Keap1*, kelch like ECH associated protein 1; *NQO1*, NAD(P)H quinone dehydrogenase 1; *GCLC*, glutamate-cysteine ligase catalytic subunit; *GCLM*, glutamate-cysteine ligase modifier subunit; *HO-1*, heme oxygenase-1; *SOD1*, superoxide dismutase 1; *ZO-1*, tight junction protein 1; *CAT*, catalase.

**Table 2 animals-10-00041-t002:** Effects of curcumin on antioxidant indexes of Jejunal mucosa in growing pigs.

Item	NBW	IUGR	IUGR + CUR
MDA (nmol/mgprot)	1.57 ± 0.09	2.01 ± 0.15 *	1.23 ± 0.13 ^#^
GPx (U/mgprot)	147.88 ± 24.69	123.20 ± 13.31	150.52 ± 31.86
T-SOD (U/mgprot)	16.54 ± 2.28	15.82 ± 3.04	28.65 ± 4.83 *^,#^
T-AOC (U/mgprot)	1.58 ± 0.15	1.62 ± 0.08	1.92 ± 0.13

Effects of curcumin on antioxidant-related indexes: malondialdehyde (MDA), glutathione peroxide enzyme (GPx), total superoxide dismutase (T-SOD) and total antioxidant capacity (T-AOC) in jejunal mucosa. All values are means ± SEM, n = 6; SEM, standard error of the mean. * *p* < 0.05 compared with the NBW group. ^#^
*p* < 0.05 compared with the IUGR group.

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
