# Peer review of "Curcumin Alleviates IUGR Jejunum Damage by Increasing Antioxidant Capacity through Nrf2/Keap1 Pathway in Growing Pigs"

_animals, 2019, doi:10.3390/ani10010041_

Round 1
Reviewer 1 Report
A manuscript entitled "Curcumin alleviates IUGR jejunum damage by increasing antioxidant capacity through Nrf2/Keap1 pathway in growing pigs" by enfa Yan et al. is contained some interesting findings. However, the current content of this article doesn't reach the level of this Journal from the following points. The authors need to consider the following points;
The authors should add the positive control of antioxidants such as Vitamin C, E etc. Further, the authors should compare your sample with these positive controls. The authors should estimate the reasons why the authors select 200mg/kg dose. Thus, the authors should add the concentration dependent manner of curcumin and should discuss the relationships between the concentration of curcumin and the change of each parameters such as Nrf2/Keap1 pathway.
This state of manuscript is very crude. So, the current content of this article doesn't reach the level of this Journal.
Author Response
Comment 1: A manuscript entitled "Curcumin alleviates IUGR jejunum damage by increasing antioxidant capacity through Nrf2/Keap1 pathway in growing pigs" by enfa Yan et al. is contained some interesting findings. However, the current content of this article doesn't reach the level of this Journal from the following points. The authors need to consider the following points.
Response 1: Thank for your effort and time spent on our manuscript, and we really appreciate for your valuable comments.
We have made point to point revision according to your valuable suggestions and asked help from a native English speaker to revise our manuscript to make our language correctly. Please see the details shown in red in the new manuscript (Figures are not marked red). Your valuable comment and efforts spent on this paper are once again highly appreciated!
Comment 2: The authors should add the positive control of antioxidants such as Vitamin C, E etc. Further, the authors should compare your sample with these positive controls.
Response 2: we feel so sorry for causing puzzles to you, and thank for your valuable effort. Our experiment was to explore whether IUGR can cause jejunum damage of growing pigs and whether dietary supplementation with curcumin can restore the intestinal damage of IUGR growing pigs to the level of NBW pigs. So we chose NBW pigs as the positive control and did not choose Vitamin C or Vitamin E as the positive control.
Comment 3: The authors should estimate the reasons why the authors select 200mg/kg dose. Thus, the authors should add the concentration dependent manner of curcumin and should discuss the relationships between the concentration of curcumin and the change of each parameters such as Nrf2/Keap1 pathway.
Response 3:Thank so much for your carefully attention. We selected 200mg/kg dose based on our previous research. In our previous experiment, weaned piglets fed a basic diet were used as the control group, and pigs fed dietary supplementation with 100, 200, 300 and 400mg/kg curcumin were used as the experimental group. The results showed that dietary supplementation with 200mg/kg of curcumin had the best effect. Please see line 96-97 shown in red and Reference 18 (Lu, N.; Qiu, J.; Ying, Z.; Zheng, Y.; Wang, T.; Zhong, X. Effects of Varying Levels of Dietary Curcumin on Growth Performance,Digestibility and Serum Parameters of Weaned Piglets. J Domest Anim Ecol. 2017, 38, 30-35.).
Comment 4: This state of manuscript is very crude.
Response 4: Thank so much for your valuable comments, and we apologize for causing puzzles to you. We have carefully revised this manuscript and asked help from a native English speaker to revise our manuscript to make our language correctly.
Reviewer 2 Report
The presentation of data in the manuscript „Curcumin alleviates IUGR jejunum damage by increasing antioxidant capacity through Nrf2/Keap1 pathway in growing pigs“ is highly problematic.
The concept of making firm conclusions based on measuring some oxidative stress markers and some NRF2 targets at the level of mRNA only, is not scientifically justified.
The design of Table 1 should be improved and the statistical significance of differences among groups should be presented in a concise way. Asterisks should be used as indicators for statistical significance.
The presentation of data on Figure 2 also calls for significant improvement. First of all, the first column should be the one related to the control samples (NBW), and the second and third columns should be the IUGR samples without and with curcumin, respectively.
The level of mRNA is not a mirror picture of the protein level. Thus, concluding about the protein level of a given protein based on the transcriptional activity of the gene is not a good approach.
Among all targets explored at the level of mRNA, we see only three confirmed at the level of protein. But, the design is not good.Even if there is an increase of Nrf2, one should follow-up on its cellular distribution (cytoplasm vs. nucleus). The same counts for Nqo1, which is known to be present in the nuclear fraction.
In the discussion section, the authors claim the following: „However, dietary supplemented 200 mg/kg curcumin can increase the expression of Nrf2 mRNA and protein. Therefore, it suggests that curcumin can activate the Nrf2 pathway by promoting the nuclear translocation of Nrf2 in jejunum of IUGR growing pigs.“ In my view, the fact that the whole study relies on findings that may be (and are) only suggestive,does not add anything new. The authors should have shown Nrf2 in the nuclei.
It is interesting, although the authors make no comment, that one of Nrf2 bona fide targets, Hmox-1 (here listed as HO-1), seems to be higly unresponsive (at the level of mRNA) in curcumin fed animals.
The authors also claim the following: „Similarly, our results showed that dietary supplemented 200 mg/kg curcumin significantly reduced the bax, bcl2 and caspase mRNA expression, thereby reducing jejunum cell apoptosis“. It is simply not acceptable to estimate the rate of apoptosis based on amount of mRNA of some genes that code for proteins that are significant for apoptotic process. If there is apoptosis, it can be seen already in genomic DNA electrophoresed in 1% agarose gel. That was not done.
Since the authors clearly claim that IUGR associates with villi atrophy, they should have explored that phenomenon and tried to determine the subcellular localisation of primary targets (nrf2, nqo1) in situ, by immunohistochemistry.
I also think that the authors did not give sufficient information on some very important elements, including the type of antibodies used and the size of amplicons.
Thank you.
Author Response
Comment 1: The presentation of data in the manuscript “Curcumin alleviates IUGR jejunum damage by increasing antioxidant capacity through Nrf2/Keap1 pathway in growing pigs” is highly problematic.
Response 1: Thank for your effort and time spent on my manuscript, and we really appreciate for your valuable comments. We have made revision according to your valuable suggestions and hope my revisions and explanations could get your approval. Please see the details shown in red in the new manuscript (Figures are not marked red) . Your valuable efforts and valuable comment spent on this paper are once again highly appreciated!
Comment 2: The concept of making firm conclusions based on measuring some oxidative stress markers and some NRF2 targets at the level of mRNA only, is not scientifically justified.
Response 2: Thank so much for your kindly and valuable comment, and we really appreciate it. We realized that the conclusions were truly overconfident, the mechanisms behind the effect of curcumin on oxidative stress was complicated and need further exploration. A discussion focused only on some oxidative stress markers and some NRF2 targets at the level of mRNA seemed to be an incomplete consideration of the present results. We have revised the conclusion with your valuable suggestion, please see line 324-328. Your valuable comment and efforts spent on this paper are once again highly appreciated!
Comment 3: The design of Table 2 should be improved and the statistical significance of differences among groups should be presented in a concise way. Asterisks should be used as indicators for statistical significance.
Response 3: Thank so much for your valuable comment and suggestion. this point has been revised, please see Table 2.
Comment 4: The presentation of data on Figure 2 also calls for significant improvement. First of all, the first column should be the one related to the control samples (NBW), and the second and third columns should be the IUGR samples without and with curcumin, respectively.
Response 4: Thank so much for your valuable comment, this point has been revised. Please see Figure 2.
Comment 5: The level of mRNA is not a mirror picture of the protein level. Thus, concluding about the protein level of a given protein based on the transcriptional activity of the gene is not a good approach.
Response 5: Thank so much for your valuable comment, we quite agree with you. protein level of a given protein based on the transcriptional activity of the gene is not appropriate. We have noticed this, Thank you very much again!
Comment 6: Among all targets explored at the level of mRNA, we see only three confirmed at the level of protein. But, the design is not good. Even if there is an increase of Nrf2, one should follow-up on its cellular distribution (cytoplasm vs. nucleus). The same counts for Nqo1, which is known to be present in the nuclear fraction.
Response 6: Thank so much for your valuable comment and suggestion. We realized that it was not sufficient to measure only three indicators at the protein level, and due to technical limitations, we also measured NRF2 and NQO1 in whole cells. In theory, it is more convincing to measure NRF2 and NQO1 in cytoplasm and nucleus. Thank you for your suggestion. In future experiments, we will try our best to measure NRF2 and NQO1 in cytoplasm and nucleus.
Comment 7: In the discussion section, the authors claim the following: “However, dietary supplemented 200 mg/kg curcumin can increase the expression of Nrf2 mRNA and protein. Therefore, it suggests that curcumin can activate the Nrf2 pathway by promoting the nuclear translocation of Nrf2 in jejunum of IUGR growing pigs.” In my view, the fact that the whole study relies on findings that may be (and are) only suggestive, does not add anything new. The authors should have shown Nrf2 in the nuclei.
Response 7: Thank so much for your carefully attention, we have revised discussion section, please see line 253-255.
Comment 8: It is interesting, although the authors make no comment, that one of Nrf2 bona fide targets, Hmox-1 (here listed as HO-1), seems to be highly unresponsive (at the level of mRNA) in curcumin fed animals.
Response 8: Thank so much for your valuable comment. Indeed, compared with the IUGR group, the mRNA expression of HO-1 in the IUGR + CUR group did not increase significantly, but only slightly increased. We speculate that the expression of HO-1 may increase significantly at the protein level.
Comment 9: The authors also claim the following: “Similarly, our results showed that dietary supplemented 200 mg/kg curcumin significantly reduced the bax, bcl2 and caspase mRNA expression, thereby reducing jejunum cell apoptosis”. It is simply not acceptable to estimate the rate of apoptosis based on amount of mRNA of some genes that code for proteins that are significant for apoptotic process. If there is apoptosis, it can be seen already in genomic DNA electrophoresed in 1% agarose gel.
Response 9: We feel so sorry for causing puzzles to you and thank so much for your valuable comment. The following sentence :“Similarly, our results showed that dietary supplemented 200 mg/kg curcumin significantly reduced the bax, bcl2 and caspase mRNA expression, thereby reducing jejunum cell apoptosis” is indeed not accurate and overconfident. We have revised it, please see line304-307. We are very grateful for your valuable comment and effort spent on this paper, and we really appreciated for it!
Comment 10: Since the authors clearly claim that IUGR associates with villi atrophy, they should have explored that phenomenon and tried to determine the subcellular localisation of primary targets (nrf2, nqo1) in situ, by immunohistochemistry.
Response 10: Thank so much for your carefully attention, previous research in our laboratory shows that IUGR associates with villi atrophy. Please see the reference: Hao Zhang, Yue Li, Yueping Chen, et al. Effects of dietary methionine supplementation on growth performance, intestinal morphology, antioxidant capacity and immune function in intra‐uterine growth‐retarded suckling piglets. Journal of Animal Physiology and Animal Nutrition, 2019. Indeed, that's really not the right way to write it, we think this part maybe not suitable for our manuscript, because our manuscript manly discussion is the effect of curcumin on the antioxidative function of IUGR jejunum. We have deleted this section. Thank you very much again!
Comment 11: I also think that the authors did not give sufficient information on some very important elements, including the type of antibodies used and the size of amplicons.
Response 11: Thank you so much for your valuable suggestion, we have supplemented the type of antibodies used and the size of amplicons. please see line 152-155 and Table 1.

Reviewer 3 Report
Animals (nov 2019).
Yan E, et al. Curcumin alleviates IUGR jejunum damage by increasing antioxidant capacity through Nrf2/Keap1 pathway in growing pigs.
The authors studied the effects of bifunctional antioxidant curcumin supplementation on the antioxidant, immunization/inflammation, apoptotic and intestinal tight junction proteins in jejunum of piglets with intrauterine growth restriction (IUGR).
The authors conclude that the addition of curcumin to the diet alleviates the jejunum oxidative stress induced by IUGR in growing of pigs improving the antioxidant and immune functions as well as the jejunal tight junctions.
The methodology involved the use of 36 female piglets of 26 days of age that were divided in 3 groups:
Normal birth weight (NBW) (n=12), Intrauterine growth restriction (IUGR) (n=12), IUGR+curcumin (IUGR+CUR) (n=12),
NBW and IUGR groups were fed a diet without curcumin and IUGR+CUR group was fed with 200mg/kg curcumin. Diets were administrated until 115 days of age.
The evaluated parameters were:
Antioxidants status (superoxide dismutase and glutathione peroxidase activities, total antioxidant capacity, and malondialdehyde content).
mRNA expression of Nrf2, GCLC, SOD1, GCLM and NQO1, protein levels of Nrf2 and NQO1,
mRNA expression of: β-actin, beta-actin;
Oxidative stress: hsp70, heat-shock protein 70;
For cell cycle: p53, tumor protein p53;
For adhesion proteins: ocln, occludin; zo-1, tight junction protein 1;
For apoptosis:
Bcl-2, B-cell CLL/lymphoma 2; caspase 3, cysteinyl aspartate specific proteinase 3;
caspase 9, cysteinyl aspartate specific proteinase 9; bax, BCL2-associated X protein;
For inflammation/immune function: IFNγ, Interferon gamma; IL-2, interleukin-2; TNF-α, tumor necrosis factor-α; IL-6, interleukin-6; IL-1β, interleukin-1β;
For antioxidant battery: Nrf2, NF-E2-related factor 2; Keap1, kelch like ECH associated protein 1; NQO1, NAD(P)H quinone dehydrogenase 1; GCLC, glutamate-cysteine ligase catalytic subunit;
GCLM, glutamate-cysteine ligase modifier subunit; HO-1, heme oxygenase-1; SOD1, superoxide dismutase 1; CAT, catalase.
The manuscript is clear but has some fails in grammar and orthography.
As for the introduction, this gives the guideline to establish the objective of the work because considers the complications associated to oxidative stress in the piglets with intrauterine growth restriction and the possibility that curcumin improves the mentioned complications.
The methods used for meet the objectives were adequate, however is necessary some observations related to this are indicated in the major concerns cited below.
OBSERVATIONS.
MAJOR CONCERNS.
Yan´s group speculates that curcumin may regulate jejunum function in IUGR growing pigs through the Nrf2/Keap1 signaling pathway. However, the jejunum's function was not measured, not at least its physiological function of digestion and absorption.
It is not indicated how the authors made sure the piglets have intrauterine growth restriction.
The source of curcumin is not indicated either.
Six piglets were euthanized at the end of experiment but it is not indicated what was done to the remaining 30 animals. It means that only 2 pigs for each group are euthanized? For statistic analyses n = 2?
For the measurement of total antioxidant capacity, please specify the principle of kit used (A015-1-1) since there are many methods focused to this parameter (ABTS, DPPH, CUPRAC, FRAP, ORAC, etc.).
Figure 2. The NBW group must be set as the first in the graph.
Figures 3, 4, 5 and 6. The NBW bars must be set as the first for each determination.
MINOR CONCERNS.
The abbreviations must be used at the first time that is required (e.g. lines 139, 91) and the abbreviation set after the complete name (e.g. lines 142,147).
There are missing spaces: line 58, line 60
-(Abstract, 26) Recommend defining NBW, IUGR.
Be more careful with grammar and orthography. The following changes are advised, though, they may not be the only ones:
-(In general) Instead of “Tremendous” use “Recent”
-(In general) Instead of “feed additive” use “food additive”
-(Simple summary, 15) “not only have” to “not only has”
-(Abstract, 39) Eliminate “which” from “which through Nrf2/Keap1”
-(Introduction, 49) “evidences showed” to “evidence showing”
-(Introduction, 51) “immune organs” to “immune organ”
-(Introduction, 57) “decreased the activities” to “decreased activities”
-(Introduction, 57) “increased the levels” to “increased levels”
-(Introduction, 59) “previous research shows an imbalance of antioxidant…” to “previous research showed that an imbalance of the antioxidant… ”
-(Introduction, 66) “(Curcuma longa)” to italics “(Curcuma longa)”
-(Introduction, 68) “curcumin, the major…” to “curcumin. The major…”
-(Introduction, 77) Please define ROS.
-(Introduction, 79) “research suggest” to “research suggested”
-(Introduction, 81) “that curcumin may be through the … pathway to regulate…” to “that curcumin may, through the … pathway, regulate…”
-(Materials and methods, 97) Correct the cite form for reference 18.
-(Materials and methods, 98) “and free to obtain water and diet, we clean the pig house… and clean the drinking…” to “and pigs had water and food ad libitum. We cleaned the pig house… and cleaned the drinking…”
-(Materials and methods, 105) “samples was” to “samples were”
-(Materials and methods, 108) “and the activities of total antioxidant...” to “and total antioxidant…”
-(Materials and methods, 124) Sentence repeated below: “The sequence of primers…”
-(Material and methods, 132) correct “occudin”.
-(Materials and methods, 143) “then denatured at 99°C” to “then protein was denatured at 99°C””
-(Materials and methods, 161) “expressed by mean… difference was significant” to “expressed as mean… difference was considered significant”
-(Results, 167) “significant difference” to “significantly different”
-(Results, 183) “dietary supplemented with…” to “the group supplemented with….”
-(Results, 192) “hsp70 as compared with…” to “hsp70 compared with…”
-(Results, 203,213,225,315) “dietary supplemented with…” to “dietary supplementation with…”
-(Results, 212) “pigs in IUGR group” to “pigs in the IUGR group”
-(Results, 217,228) “no significant among IUGR group,… IUGR+CUR group” to “no significant differences among the IUGR, NBW and IUGR+CUR groups”
-(Results, 231). Figure 6. Correct “tublin”.
-(Discussion, 245) “make the activity” to “maintain the activity”
-(Discussion, 249) “such stresses” to “such stressors”
-(Discussion, 268) “anti-apoptosis and…” to “anti-apoptosis role and…”
-(Discussion, 271) “life [32], our…” to “life [32]. Our…”
-(Discussion, 278) “cells and inflammatory due…” refers to expression during inflammation or also in inflammation?
-(Discussion, 285) “Previous study have shown” to “A previous study has shown”
-(Discussion, 292) “In a model… apoptosis, Nrf2-deficient…” to “In a model… apoptosis, it was demonstrated that Nrf2-deficient…”
-(Discussion, 307) “important connections” to “important connection”
Reformulate the following phrases, as they are not clear:
-(Abstract, 47) “IUGR embodied in the newborn fetal birth weight below…”
-(Materials and methods, 148) “then the membrane was transferred for transfer”
-(Discussion, 278) “The TNFα were continuously increased…”
-(Discussion, 302) “In a rat model of jejunum injury…”
-(Conclusions, 328) “helpful in exploit a new…” to “helpful in exploiting a new…”
Author Response
Comment 1: The manuscript is clear but has some fails in grammar and orthography. As for the introduction, this gives the guideline to establish the objective of the work because considers the complications associated to oxidative stress in the piglets with intrauterine growth restriction and the possibility that curcumin improves the mentioned complications. The methods used for meet the objectives were adequate, however is necessary some observations related to this are indicated in the major concerns cited below.
Response 1: Thank for your effort and time spent on my manuscript, and we really appreciate for your precious recognition of this paper and valuable comments. We have made point to point revision according to your valuable suggestions and hope my revisions and explanations could get your approval. Please see the details shown in red in the new manuscript (Figures are not marked red). Your valuable efforts and valuable comment spent on this paper are once again highly appreciated!
Comment 2: the jejunum's function was not measured, not at least its physiological function of digestion and absorption.
Response 2: We feel so sorry for causing puzzles to you and thank so much for your valuable comment. We have previously measured the activity of jejunal sucrase, lactase, maltidase. Because this manuscript mainly discusses the antioxidant capacity of curcumin, the determination results of disaccharidase are not included in this article. The results of the disaccharidase assay are shown in the table below.
|
|
NBW |
IUGR |
IUGR+CUR |
p Value |
|
Maltase(U/mgprot) |
53.44±4.82ab |
39.12±5.56b |
59.57±4.90a |
P=0.035 |
|
Sucrase(U/mgprot) |
68.47±10.99 |
68.09±10.44 |
103.98±12.34 |
P=0.063 |
|
Lactase(U/mgprot) |
62.17±4.96a |
38.31±5.22b |
56.39±5.50a |
P=0.015 |
All values are means ± SEM, n=6; SEM, standard error of the mean. In the same row, values with different letter superscripts are significant difference (P<0.05), and with same letter superscripts are no significant difference (P>0.05).
Comment 3: It is not indicated how the authors made sure the piglets have intrauterine growth restriction.
Response 3:
Thank so much for your valuable comment and effort spent on this paper. A piglet was defined as intrauterine growth restriction when its birth weight was two standard deviations below the mean birth weight of the total population. Please see line 47-49.
Comment 4: The source of curcumin is not indicated either.
Response 4:Thank you so much for your valuable suggestion, Curcumin provided by Guangdong Kehu Biotechnology Research and Development Center, purity ≥98%. Please see line 96.
Comment 5: Six piglets were euthanized at the end of experiment but it is not indicated what was done to the remaining 30 animals. It means that only 2 pigs for each group are euthanized? For statistic analyses n = 2?
Response 5: Thank so much for your valuable comment and effort spent on this paper. We have revised a clear statement that six piglets were selected from each group and euthanized at the end of experiment. Please see line 101.
Comment 6: For the measurement of total antioxidant capacity, please specify the principle of kit used (A015-1-1) since there are many methods focused to this parameter (ABTS, DPPH, CUPRAC, FRAP, ORAC, etc.).
Response 6: we feel so sorry for causing puzzles to you, and thank for your valuable effort, the measure method of T-AOC kit (A015-1-1) is FRAP. Please see line 109.
Comment 7: Figure 2. The NBW group must be set as the first in the graph.
Response 7: Thank so much for your valuable comment, this point has been revised. Please see Figure 2.
Comment 8: Figures 3, 4, 5 and 6. The NBW bars must be set as the first for each determination.
Response 8: Thank so much for your carefully attention, this point has been revised, please see Figure 3, 4, 5 and 6.
Comment 9: The abbreviations must be used at the first time that is required (e.g. lines 139, 91) and the abbreviation set after the complete name (e.g. lines 142,147).
Response 9: Thank so much for your carefully attention, this point has been revised.
Comment 10: There are missing spaces: line 58, line 60.
Response 10: Thank so much for your carefully attention, this point has been revised.
Comment 11: (Abstract, 26) Recommend defining NBW, IUGR.
Response 11: Thank you so much for your valuable suggestion, this point has been revised. please see line 26-27.
Comment 12: (In general) Instead of “Tremendous” use “Recent”. (In general) Instead of “feed additive” use “food additive”.
Response 12: Thank so much for your valuable suggestions, this point has been revised.
Comment 13: (Simple summary, 15) “not only have” to “not only has”
Response 13: Thank so much for your carefully attention, this point has been revised, please see line 15.
Comment 14: (Abstract, 39) Eliminate “which” from “which through Nrf2/Keap1”
Response 14: Thank so much for your carefully attention, this point has been revised, please see line 40-41.
Comment 15: (Introduction, 49) “evidences showed” to “evidence showing”
Response 15: Thank so much for your carefully attention, this point has been revised, please see line 50.
Comment 16: (Introduction, 51) “immune organs” to “immune organ”
Response 16: Thank so much for your carefully attention, this point has been revised, please see line 52.
Comment 17: (Introduction, 57) “decreased the activities” to “decreased activities”
Response 17: Thank so much for your valuable comment, this point has been revised, please see line 58.
Comment 18: “increased the levels” to “increased levels”
Response 18: Thank so much for your carefully attention, this point has been revised, please see line 58.
Comment 19: (Introduction, 59) “previous research shows an imbalance of antioxidant…” to “previous research showed that an imbalance of the antioxidant… ”
Response 19: Thank so much for your valuable comment, this point has been revised, and replaced “showed” with “proved”, please see line 60.
Comment 20: (Introduction, 66) “(Curcuma longa)” to italics “(Curcuma longa)”
Response 20: We are so sorry for causing puzzles to you, positive effect means good effect, and this point has been revised, please see line 66.
Comment 21: (Introduction, 68) “curcumin, the major…” to “curcumin. The major…”
Response 21: Thank so much for your valuable comment, this point has been revised, please see line 68.
Comment 22: (Introduction, 77) Please define ROS.
Response 22: Thank for your valuable suggestion, this point has been revised, please see line 77.
Comment 23: (Introduction, 79) “research suggest” to “research suggested”
Response 23: We feel so sorry for causing puzzles to you and thank so much for your valuable comment. this point has been revised, please see line 79.
Comment 24: (Introduction, 81) “that curcumin may be through the … pathway to regulate…” to “that curcumin may, through the … pathway, regulate…”
Response 24: Thank so much for your carefully attention, this point has been revised, please see line 81-83.
Comment 25: (Materials and methods, 97) Correct the cite form for reference 18.
Response 25: Thank for your valuable suggestion, this point has been revised, please see line 96-97.
Comment 26: (Materials and methods, 98) “and free to obtain water and diet, we clean the pig house… and clean the drinking…” to “and pigs had water and food ad libitum. We cleaned the pig house… and cleaned the drinking…”
Response 26: Thank for your carefully attention, this point has been revised, please see line 98-100.
Comment 27: (Materials and methods, 105) “samples was” to “samples were”
Response 27: Thank so much for your carefully attention, this point has been revised, please see line 105.
Comment 28: (Materials and methods, 108) “and the activities of total antioxidant...” to “and total antioxidant…”
Response 28: Thank so much for your kindly and valuable comment, this point has been revised, please see line 108-109.
Comment 29: (Materials and methods, 124) Sentence repeated below: “The sequence of primers…”
Response 29: Thank so much for your kindly and valuable comment, and we really appreciate it. this point has been revised, please see line 124-125.
Comment 30: (Material and methods, 132) correct “occudin”
Response 30: Thank so much for your carefully attention, this point has been revised, please see line 133.
Comment 31: (Materials and methods, 143) “then denatured at 99°C” to “then protein was denatured at 99°C””
Response 31: Thank so much for your valuable comment, this point has been revised, please see line 144-145.
Comment 32: (Materials and methods, 161) “expressed by mean… difference was significant” to “expressed as mean… difference was considered significant”
Response 32: Thank for your valuable suggestion, this point has been revised, please see line 163-164.
Comment 33: (Results, 167) “significant difference” to “significantly different”
Response 33: Thank so much for your kindly and valuable comment, this point has been revised.
Comment 34: (Results, 183) “dietary supplemented with…” to “the group supplemented with….”
Response 34: Thank so much for your carefully attention, this point has been revised, please see line 184.
Comment 35: (Results, 192) “hsp70 as compared with…” to “hsp70 compared with…”
Response 35: Thank so much for your kindly and valuable comment, and we really appreciate it. This point has been revised, please see line 193-194.
Comment 36: (Results, 203,213,225,315) “dietary supplemented with…” to “dietary supplementation with…”
Response 36: We feel so sorry for causing puzzles to you and thank so much for your valuable comment. This point has been revised.
Comment 37: (Results, 212) “pigs in IUGR group” to “pigs in the IUGR group”
Response 37: Thank for your valuable suggestion, this point has been revised, please see line 213.
Comment 38: (Results, 217,228) “no significant among IUGR group,… IUGR+CUR group” to “no significant differences among the IUGR, NBW and IUGR+CUR groups”
Response 38: Thank so much for your kindly and valuable comment, this point has been revised, please see line 218, 229.
Comment 39: (Results, 231). Figure 6. Correct “tublin”
Response 39: Thank so much for your carefully attention, this point has been revised, please see figure 6.
Comment 40: (Discussion, 245) “make the activity” to “maintain the activity”
Response 40: Thank so much for your kindly and valuable comment, and we really appreciate it. This point has been revised, please see line 246.
Comment 41: (Discussion, 249) “such stresses” to “such stressors”
Response 41: Thank so much for your carefully attention, this point has been revised, please see line 250.
Comment 42: (Discussion, 268) “anti-apoptosis and…” to “anti-apoptosis role and…”
Response 42: Thank for your valuable suggestion, this point has been revised, please see line 268.
Comment 43: (Discussion, 271) “life [32], our…” to “life [32]. Our…”
Response 43: Thank so much for your carefully attention, this point has been revised, please see line 271.
Comment 44: (Discussion, 278) “cells and inflammatory due…” refers to expression during inflammation or also in inflammation?
Response 44: Thank so much for your kindly and valuable comment, it is mean think if the expression level of Nrf2 is reduced, inflammation will occur.
Comment 45: (Discussion, 285) “Previous study have shown” to “A previous study has shown”
Response 45: Thank so much for your carefully attention, this point has been revised, please see line 285.
Comment 46: (Discussion, 292) “In a model… apoptosis, Nrf2-deficient…” to “In a model… apoptosis, it was demonstrated that Nrf2-deficient…”
Response 46: Thank for your valuable suggestion, this point has been revised, please see line 292.
Comment 47: (Discussion, 307) “important connections” to “important connection”
Response 47: Thank so much for your kindly and valuable comment, this point has been revised.
Comment 48: Reformulate the following phrases, as they are not clear:
-(Abstract, 47) “IUGR embodied in the newborn fetal birth weight below…”
-(Materials and methods, 148) “then the membrane was transferred for transfer”
-(Discussion, 278) “The TNFα were continuously increased…”
-(Discussion, 302) “In a rat model of jejunum injury…”
-(Conclusions, 328) “helpful in exploit a new…” to “helpful in exploiting a new…”
Response 48: We feel so sorry for causing puzzles to you and thank so much for your valuable comment. this point has been revised, please see line 47-48, 150, 279, 303, 329.

Round 2
Reviewer 1 Report
Thanks, you made plenty of changes and answered all my comments.
Now the manuscript is more understandable.
I recommend the manuscript to be accepted.
Reviewer 2 Report
Dear authors,
I have carefully read explanations offered in you reply.
Please, note that already in abstract, the explanation for IUGR was written twice.
All the best,
Author Response
Please see the attachment.

This manuscript is a resubmission of an earlier submission. The following is a list of the peer review reports and author responses from that submission.
Round 1
Reviewer 1 Report
A manuscript entitled "Curcumin alleviates IUGR jejunum damage by increasing antioxidant capacity through Nrf2/Keap1 pathway in growing pigs" by enfa Yan et al. is contained some interesting findings. However, the current content of this article doesn't reach the level of this Journal from the following points. The authors need to consider the following points;
The authors should add the positive control of antioxidants such as Vitamin C, E etc. Further, the authors should compare your sample with these positive controls. The authors should estimate the reasons why the authors select 200mg/kg dose. Thus, the authors should add the concentration dependent manner of curcumin and should discuss the relationships between the concentration of curcumin and the change of each parameters such as Nrf2/Keap1 pathway.
This state of manuscript is very crude. So, the current content of this article doesn't reach the level of this Journal.
Reviewer 2 Report
The presentation of data in the manuscript „Curcumin alleviates IUGR jejunum damage by increasing antioxidant capacity through Nrf2/Keap1 pathway in growing pigs“ is highly problematic.
The concept of making firm conclusions based on measuring some oxidative stress markers and some NRF2 targets at the level of mRNA only, is not scientifically justified.
The design of Table 1 should be improved and the statistical significance of differences among groups should be presented in a concise way. The presentation that includes superscript letters is not the solution for this kind of presentation. I would also like to see whether there is a level of statistical difference £0.001.
The presentation of data on Figure 2 also calls for significant improvement. First of all, the first column should be the one related to the control samples (NBW), and the second and third columns should be the IUGR samples without and with curcumin, respectively. Interpretation of Fig. 1 is strange: „IUGR pigs had lower mRNA expression of Nrf2, Sod1, Gclc and Nqo1 as compared with NBW pigs (P<0.05).“ What I see is a higher level of Nqo1 and Cat transcripts (not statistically significant, but still obviously higher) in IUGR than in the controls. After carefuly looking at the graphics, I have the impression that Keap1, Sod and Gclc are not normalized to the value „1“ – they seem to be lower.
Figure 3 clearly indicates that inclusion of one more group of animals, NBW fed with curcumin, would significantly improve the whole study.
Most targets presented on Figure 4 are associated with inflammation.They represent the cluster of inflammation-related genes.
When I compare Figs. 1 and 5, I see an increased level of Nqo1 mRNA in IUGR/IUGR+CUR (compared to NBW) but that is not visible at the level of protein where we see a dramatic increase of the Nqo1 protein in CUR-treated. It is also clearly visible a significantly smaller amount of Nqo1 protein in IUGR. How can that be explained?
Clearly, the level of mRNA is not a mirror picture of the protein level. Thus, concluding about the protein level of a given protein based on the transcriptional activity of the gene is not a good approach.
Among all targets explored at the level of mRNA, we see only three confirmed at the level of protein. But, the design is not good.Even if there is an increase of Nrf2, one should follow-up on its cellular distribution (cytoplasm vs. nucleus). The same counts for Nqo1, which is known to be present in the nuclear fraction.
In the discussion section, the authors claim the following: „However, dietary supplemented 200 mg/kg curcumin can increase the expression of Nrf2 mRNA and protein. Therefore, it suggests that curcumin can activate the Nrf2 pathway by promoting the nuclear translocation of Nrf2 in jejunum of IUGR growing pigs.“ In my view, the fact that the whole study relies on findings that may be (and are) only suggestive,does not add anything new. The authors should have shown Nrf2 in the nuclei.
It is interesting, although the authors make no comment, that one of Nrf2 bona fide targets, Hmox-1 (here listed as HO-1), seems to be higly unresponsive (at the level of mRNA) in curcumin fed animals. The authors should be aware that curcumin may and may not increase expression of hsp70. For example, in leukemic cells, curcumin „induces nuclear translocation of the heat shock transcription factor (HSF)-1, its binding to a heat shock regulatory element (HSE), and the subsequent activation of the hsp70 promoter...“ (Cancer Lett. 2009 Jul 8;279(2):145-54. doi: 10.1016/j.canlet.2009.01.031.). The question that remained unanswered here is: What is the status of the hsp70 PROTEIN?
The authors also claim the following: „Similarly, our results showed that dietary supplemented 200 mg/kg curcumin significantly reduced the bax, bcl2 and caspase mRNA expression, thereby reducing jejunum cell apoptosis“. It is simply not acceptable to estimate the rate of apoptosis based on amount of mRNA of some genes that code for proteins that are significant for apoptotic process. If there is apoptosis, it can be seen already in genomic DNA electrophoresed in 1% agarose gel. That was not done.
The authors claim that: „In our study, curcumin can significantly inhibit the TNF-α, INF-γ and IL-6 mRNA expression, and increase IL-2 mRNA expression in jejunal mucosa, which are similar to xun Wenjuan team's conclusions [36]. Therefore, we speculate that curcumin may alleviates jejunal inflammation in IUGR growing pigs through the Nrf2 pathway“, My impression is that the decrease of activity of axis, which includes TNF-α and IL-6 mRNA, represents the consequence of a strong inhibition of NF-kappaB signaling pathway (COX-2 status was not determined) and, based on hmox-1 and nqo1 status, only selective and moderate modulation of nrf2-related targets.
Since the authors clearly claim that IUGR associates with villi atrophy, they should have explored that phenomenon and tried to determine the subcellular localisation of primary targets (nrf2, nqo1) in situ, by immunohistochemistry.
I also think that the authors did not give sufficient information on some very important elements, including the type of antibodies used. I appreciate the detailed list of primers. Of note, if one wants to make an accurate quantification with SybrGreen, the length of amplicons should not exceed 500 bps. Ideally, the amplicons shuould be between 100 and 150 bps. The authors did not offer the length of amplicons, so I made some checks myself: the amplicons that I checked significantly varied in length: from 108 bps (caspase 9) up to 565 bps (occludin). I am concerned that the design of some primers makes quantification of transcripts very inaccurate. Of note, I could not find the binding site for the revers primer IFNG.
Thank you.